# The Spatial Distribution of Crimean–Congo Haemorrhagic Fever and Its Potential Vectors in Europe and Beyond

**DOI:** 10.3390/insects14090771

**Published:** 2023-09-17

**Authors:** Jane Paula Messina, G. R. William Wint

**Affiliations:** 1School of Geography and the Environment, University of Oxford, S. Parks Rd., Oxford OX1 3QY, UK; 2Oxford School of Global and Area Studies, University of Oxford, 13 Bevington Rd., Oxford OX2 6LH, UK; 3Environmental Research Group, Department of Biology, University of Oxford, 11a Mansfield Rd., Oxford OX1 3SZ, UK; william.wint@gmail.com

**Keywords:** Crimean–Congo haemorrhagic fever, CCHF, disease risk mapping, ticks, *Hyalomma*, global health, spatial modelling

## Abstract

**Simple Summary:**

Crimean–Congo haemorrhagic fever (CCHF) is an emerging disease that is spreading across the globe. We originally released a predictive map in 2015, which we now update and improve by extending its coverage to Europe and by incorporating the limiting distributions of its main confirmed or potential tick vectors in Europe: *Hyalomma marginatum* and *Hyalomma lusitanicum*. Our new models suggest that the disease risk is now predicted to cover substantial parts of southern and central Europe. These maps can be used to more effectively target surveillance and monitoring.

**Abstract:**

Crimean–Congo haemorrhagic fever (CCHF) is considered to be spreading across the globe, with many countries reporting new human CCHF cases in recent decades including Georgia, Türkiye, Albania, and, most recently, Spain. We update a human CCHF distribution map produced in 2015 to include global disease occurrence records to June 2022, and we include the recent records for Europe. The predicted distributions are based on long-established spatial modelling methods and are extended to include all European countries and the surrounding areas. The map produced shows the environmental suitability for the disease, taking into account the distribution of the most important known and potential tick vectors *Hyalomma marginatum* and *Hyalomma lusitanicum,* without which the disease cannot occur. This limits the disease’s predicted distribution to the Iberian Peninsula, the Mediterranean seaboard, along with Türkiye and the Caucasus, with a more patchy suitability predicted for inland Greece, the southern Balkans, and extending north to north-west France and central Europe. These updated CCHF maps can be used to identify the areas with the highest probability of disease and to therefore target areas where mitigation measures should currently be focused.

## 1. Introduction

Crimean–Congo haemorrhagic fever (CCHF) is a viral disease of the Nairovirus genus (family Bunyaviridae) transmitted by ticks. It was first identified in Crimea in 1944 [1,2] and then later found to be the same virus causing disease outbreaks in the Congo river basin [3,4]. Considered to be a globally emerging infectious disease, many countries have only submitted reports of human infection in the last 20 years, including Albania (2001) [5], Türkiye (2002) [6], Georgia (2009) [7], and Spain (2013) [8]. CCHF in humans has also recently occurred after an extensive absence, in places such as south-west Russia [9] and Central Africa [10].

The disease is transmitted to humans by ticks via wild and domesticated animal reservoirs, which themselves do not display symptomatic disease [11]. No CCHF-specific antiviral drug or vaccine currently exists for animals or humans. Several tick species can carry CCHF virus (CCHFV), with members of the genus *Hyalomma* considered as the primary vectors. As such, although human CCHFV infection is uncommon, those living or working in close proximity to livestock or vector habitats are at greater risk for infection, and fatality can be as high as 40% [1]. Whilst there are a number of different CCHF vector species within the global range of the disease, only two are widespread in Europe [12]. These are *H. marginatum*, which is found throughout southern Europe, the Middle East, and North Africa as far east as India; and *H. lusitanicum*, which is limited to southwestern Europe and neighbouring parts of North Africa [13,14,15]). It should be noted that both these ticks are adapted to hot and dry or semiarid environments [1,16,17,18,19].

The distribution of CCHF was modelled globally in 2015 by Messina et al. [20], a study that predicted the disease to be possible in parts of eastern and southern Europe. Whilst the 2015 model successfully predicted that Spain was suitable for CCHF, the published outputs were adjusted so that ecological suitability for CCHF was only shown in countries that had reported cases. The 2015 model also did not incorporate vector distributions in the disease predictions. The 2015 model did not include Spain, where the first case was reported in 2016 [21], and which was not therefore shown as a potential area for CCHF distribution. In addition, CCHF predictions from other authors have become available [22].

Here, we re-estimate CCHF distributions globally, with the update taking into account more recent disease occurrence data. The distribution of its primary vectors was also modelled for Europe and its neighbouring areas and used for masking the final results in this region. Two sets of spatial modelling were performed using separate predictor data suites and well-established spatial modelling techniques for disease risk mapping. These models were (a) CCHF itself in humans and (b) the two associated vector species. The vector distribution models allow for refined mapping of CCHF ecological suitability compared to 2015 by masking out those areas in the basic CCHF spatial model where no vectors are present.

## 2. Materials and Methods

The overall modelling aim was to establish a statistical relationship between known presence (or absence) and the values of a series of selected predictor covariates. These relationships were calculated for a set of sample locations, and the estimated equations were then applied to maps of the covariates that provide values at a pixel resolution for the entire area of interest. This resulted in a modelled spatial distribution showing the probability of presence at the resolution of the covariate maps—which were standardised in this project at 1 square kilometre. The models were produced with the specific objective of providing disease suitability assessments for Europe and its neighbouring areas as far east as the Caspian Sea and including north Africa.

### 2.1. Vector Suitability Mapping

The vectors were each modelled for their entire reported ranges plus a buffer zone (an area around the range boundary) of at least 200 km wide. The vector spatial distribution modelling was performed using both Random Forest (RF) and Boosted Regression Trees (BRT) implemented through the VECMAP^®^ Software Suite v1 (AVIA-GIS, Zoersel, Belgium), to model presence and absence, producing estimates of the probability of presence. Five replicates of each method, with a 25% holdback, were run, and the results were combined to produce ensemble mean, median, minimum, and maximum predictions of the probability of presence. The combination of methods tends to reduce the tendency for BRT to overfit, especially for training data covering relatively restricted areas such as that available for *H. lusitanicum.*

These methods require approximately equal numbers of presence and absence points to be offered to each modelling run. The occurrence data were obtained primarily from VectorNet [15] shown in Appendix A, supplemented by approximately 60 records for each species from the Global Biodiversity Information Facility (GBIF, https://www.gbif.org, accessed on 15 October 2022). The VectorNet data consisted of both point and polygon data, differentiated into present, absent, and introduced categories. This last category represents records of temporary presence—often from migratory birds—and are not indicative of established populations. These records were therefore discarded. Five points were defined for each polygon and assigned as present or absent according to the polygon status, and to these, any point data from either VectorNet or GBIF were added.

A location was defined as suitable for each vector as indicated in Appendix A and mapped in Appendix A. These were taken from a number of published sources [14,23,24,25,26,27,28,29,30]. Whilst both species do well in most woodland, grassland, shrubland, and cropland environments, only *H. lusitanicum* occupies dense woodland and only *H. marginatum* is associated with sparse vegetation. Though *H. marginatum* does require minimum temperatures through the summer and needs relatively moist conditions, *H. lusitanicum* is able to occupy areas with hotter and drier summer conditions and requires a relatively warm autumn. There are few if any geo-referenced occurrence records for other potential CCHFV vectors, and as such these were not incorporated into the vector modelling. Once the suitable and unsuitable areas had been defined, and following the methodology used in Wint et al. (2020) [31], for each species, pseudo-absence points were randomly assigned to unsuitable areas inside the range depicted by Kolonin (2009) [13], and within approximately 300 km^2^ of known presences. Beyond these limits, absences were assigned to all areas, irrespective of suitability.

All points were then aggregated to a 10 km grid to combine any multiple overlapping records. The number of presence and absence points was then adjusted by reducing the number of the larger class to match that of the less frequent class to produce a final balanced output dataset for spatial modelling. These are shown in Figure 1.

### 2.2. Suitability Mapping for CCHF in Humans

The distribution of the disease within Europe is somewhat restricted and the number of reported disease occurrence locations provides insufficient training data to run an effective model for the region alone. It was therefore necessary to run the disease model based on global data to inform the potential suitability in Europe.

The occurrence database compiled for Messina et al. (2015) [20] was updated using the same methodologies to include records from new peer-reviewed literature and case reports up until July 2022. As with Messina et al. (2015), an occurrence was defined as one or more laboratory-confirmed human CCHF infection(s) occurring at a unique location (the same administrative area or 1 × 1 km pixel for precise locations) within one calendar year, and administrative area polygons greater than one square degree in area (which is approximately 12,300 km^2^ at the equator) were also removed from the database, as their inclusion in niche modelling would introduce a large amount of spatial imprecision.

In total, 1437 point occurrence records and 854 polygon occurrence records (representing the years 1952–2022) were included in our models after quality control (Appendix A for global map), with the assumption that any recorded location of human CCHF occurrence during this time period would represent an environment permissible for the occurrence of human disease cases. Appendix A shows histograms for the number of CCHF occurrence locations by year of publication. Occurrence and generated pseudo-absence locations (described in greater detail below) for Europe are shown in Figure 2.

Global gridded data (of 1 km × 1 km resolution) were taken from a covariate data suite (Appendix A) for a set of four explanatory covariates. The covariates were chosen based on factors known or hypothesised to contribute to suitability for CCHFV transmission to humans based upon the national-level studies described in the introduction. These included (i) annual mean precipitation interpolated from global meteorological stations; (ii) mean land surface temperature derived from NASA’s moderate-resolution imaging spectrometer (MODIS) imagery, intended to capture the generally warm and arid climate zones where CCHFV is transmitted; (iii) 1 km resolution measure of the mean annual Enhanced Vegetation Index (EVI, also from MODIS); and (iv) the proportion of each 1 km×1 km grid cell covered by shrub or grass land cover types derived from the Earthenv consensus land cover datasets. No covariate grids were shown to be adversely affected by multicollinearity based on a Pearson’s correlation threshold of 0.8 or higher.

Having assembled occurrence and covariate datasets, BRT modelling methodology was employed to model the relationship between the probability of CCHF occurrence (as defined earlier) and the four covariates. For the occurrence points, covariate values were sampled at each occurrence location for precise locations (such as cities or towns) or averaged within each polygon-level occurrence (such as counties or provinces). As with the vector modelling, the CCHF models require not only presence data but also pseudo-absence data defining areas of potentially unsuitable environmental conditions at unsampled locations, since data on absence of human disease are rarely reported.

An evidence consensus score was derived from a series of evidence types for the presence or absence of CCHF disease in humans at the national level. This information was used to rank a country from −100 (consensus on absence) to +100 (consensus on presence) using the same methodology as described in Messina et al. [20] and more recent reports of CCHF transmission since the publication of that study (see Figure 3). Finally, to represent the environmental conditions in locations where human disease has not been reported, 5000 background points were randomly generated and weighted based on these values. More background absence points were assigned to areas with a high consensus on absence, and fewer were assigned to areas with a high consensus on presence.

BRT models were fitted to 50 separate bootstraps of the entire dataset (including both point and polygon occurrences and pseudo-absence points) to increase prediction robustness (taking the mean across all 50 BRT models for each pixel) and provide estimates of model uncertainty (taking the 1st and 3rd quartiles of the model outputs for each pixel). Each of the 50 individual models was fitted using the gbm.step subroutine in the dismo package in R. All other tuning parameters of the algorithm were retained as in Messina et al. [20]

## 3. Results

### 3.1. Vector Distributions

Predicted presence (and absence) for each vector species are shown in Figure 4. Three estimates of presence were derived from the model replicates—the mean, minimum, and maximum values, each coded as present if the calculated probability of presence value exceeded 0.5. This resulted in four distribution categories in each map: minimum, mean, and maximum predicted vector extent, and absence. All were masked by the suitability limits described in the methods section.

*H. lusitanicum* is predicted to occur throughout the Iberian Peninsula and western North Africa as well as some coastal areas of southern France and Italy. The mean predicted extent for *H. marginatum* covers most of southern Europe, the Caucasus, and Norh Africa; the maximum predicted extent stretches substantially further north into parts of France, Central Europe, and the Caucasus than do the mean predictions.

The vector models generally reflect the training data well, especially for *H. lusitanicum* where both sensitivity and specificity or the RF models average well above 0.9 and the receiver operating characteristic (ROC) values of the BRT models exceed 0.95. This is perhaps to be expected as the species range is relatively restricted and less environmentally heterogeneous than that of *H. marginatum*. The *H. marginatum* models also perform well with associated Kappa values of >0.84 for the RF models and ROC values >0.89 for the BRT models. Whilst these models have a high sensitivity (>0.83), their specificity is a little lower at 0.69–0.72, meaning that the models predicted a significant number of false negatives. These are largely to the east of the modelled area and outside the project area of interest, further discussed below.

The most frequently used predictors for *H. marginatum* are (a) the minimum values of the Normalised Difference Vegetation Index (NDVI), (b) aspects of the day and night-time temperature, (c) the timing and levels of rainfall, and (d) aspects of minimum relative humidity. For *H. lusitanicum*, the best predictors relate to (a) monthly rainfall seasonality and the degree to which it varies over the year; and (b) the levels and timing of day and night-time temperature. The importance of the top ten predictors for each vector and modelling method is given in Appendix A.

### 3.2. CCHF Distribution in Humans

Figure 5 shows the resulting mean predictions for CCHF probability of occurrence (ranging 0–1), displayed with masks of minimum, maximum, and mean predicted vector presence (where at least one of the modelled probability values for either vector’s presence was greater than or equal to 0.5), as well as the unmasked version. Appendix A shows the full global modelled distribution (beyond Europe; unmasked). Uncertainty estimates for each pixel were possible from the ensemble approach (by using the interquartile range of all 50 predictions) and are shown in Appendix A.

The proportional influence of covariates toward the probability predictions in the ensemble of models is shown in Figure 6, with the total proportion equalling 100 percent. The average Area Under the Curve (AUC) across all models was 0.74 (range 0.67–0.79). Land surface temperature was by far the greatest contributor to the models, followed by precipitation, shrub land cover, and then EVI.

The unmasked disease predictions suggest that the environmental suitability for the disease extends well into northern Europe and the northern Caucasus—far beyond the distributions of the two vector species considered. Limiting the predictions to the area covered by the vector distributions implies at best (i.e., masked with minimum vector distribution area) that Mediterranean France, Spain, Türkiye, and much of the Mediterranean seaboard are all predicted to be widely suitable for the disease, with a patchier suitability predicted for inland Greece and the southern Balkans.

The recorded disease locations (Appendix A), which extend well into the Caucasus and central Europe, matched most closely to the maximum predicted vector extents. These extents were therefore used for masking the final human disease suitability maps. This level of masking shows that the predicted disease range extends significantly further north to 47 °N (north central France) and eastward. It is also evident that the vector masking ‘removes’ the predicted disease occurrence from Israel, Palestine, Lebanon, Syria, and Iraq. These are areas for which there are no presence records of either vector, and so are also not predicted to support either vector. Because these maps are not masked by areas where the consensus layer shows (high) consensus, they show France and Italy as having a high probability of CCHF, even though there have not been any reports of cases thus far. Still, Figure 5 shows several patches of environmental suitability within the predicted vector ranges in these countries. Currently, Northern Europe is not predicted to be ecologically suitable for CCHF, due to a lack of vector presence. Appendix A shows the unmasked probability of CCHF occurrence for the full modelled extent. Uncertainty estimates for the updated CCHF probability of occurrence maps for each pixel are shown in Appendix A.

The difference between the unmasked predictions for 2022 and 2015 [20] (Appendix A) is presented in Figure 7 and compares the current predictions masked with the maximum predicted vector extent with those of 2015—with the red areas indicating where the current model predicts substantially higher ecological suitability for CCHF. This map reflects the additional CCHFV presence records included in the current models in the eastern part of the region (Appendix A). Whilst much of Europe’s predicted ecological suitability remains the same in this updated model, much of the south is predicted to be substantially more suitable: particularly pockets throughout Spain, France, Italy, Greece, the Balkans, Türkiye, and North Africa.

As emphasised above, this reflects the addition of new training data for the 2022 model and changes in the masking used from the country level (2015) to vector presences (2022). It should perhaps be noted that the two models were run at different spatial resolutions (2015 at 5 km and 2022 at 1 km) and so the differences are not strictly valid at the higher resolution, and the map in Figure 7 should only be used for reference on general areas where ecological suitability predictions are different.

Figure 8 shows areas where the masked CCHF probability of occurrence is high (greater than or equal to 0.75) within the maximum predicted presence of either suitable vector species, and, as such, may be used as a guide for prioritising future virus surveillance efforts in humans and animals.

The equivalent high-probability maps for each vector species are presented in Appendix A. It can be seen that areas bordering the Adriatic and Mediterranean Seas are likely to be particularly important for vector and disease surveillance, as well as the majority of the Iberian peninsula and Türkiye.

## 4. Discussion

This study provides updated baseline data for monitoring future changes in the distribution of CCHF and its associated vectors for Europe and its neighbouring areas. The predicted distributions suggest that a number of countries that have yet to record CCHF have areas that are environmentally suitable for the disease, especially those with Mediterranean coastlines (France, Italy, the southern Balkans, Western Asia, and North Africa). Both the unmasked predictions and the maximum predicted extents of the two vector species (which better reflect the distribution of reported human CCHF case locations than the medium extent) imply a potential CCHF distribution extending significantly northward into mainland western and central Europe. It is tempting to suggest that this reflects an increased likelihood of northward shifts in range driven by climate change. It is also worth noting that these maximum predicted extents overlap some of the VectorNet distributions assigned as ‘introduced’ (Appendix A) because they were thought to be too far beyond the current northern distribution limits to represent established populations. It could be that these introduced presence points are in fact the first indications of future spread and establishment.

Further cartographic refinements are required in order to help differentiate endemic from epidemic-prone areas, particularly in areas where there is less certainty about the presence or absence of CCHFV overall. The CCHFV occurrence database used in creating the CCHF ecological suitability map has been updated to mid-2022, and can continue to be updated with new information as it becomes available. Other possible improvements are the inclusion of other potential CCHFV vector species into the masking process (discussed below) and the addition of more data types, such as those related to CCHFV seroprevalence if more data were to become available. For the latter, recent research has elucidated potentially important CCHFV reservoir species in non-endemic areas (e.g., France, Italy) that should also be considered; for example, imported sheep and goats may be more susceptible to CCHFV infection and therefore might promote virus circulation to humans where suitable vectors are present [32,33]. These efforts to improve the mapping of current ecological suitability could further facilitate identification of potential transmission foci.

Already, these maps have been improved from the 2015 [20] versions due not only to the updated occurrence database and higher spatial resolution (now 1 km x 1 km), which includes significant new data from Türkiye, Spain, and the Caucasus, but also the inclusion of vector suitability instead of evidence consensus to mask disease predictions.

Although a similar occurrence dataset was used in the current model and that of Okely et al. (2020) [22], our predictions do not have the high ecological suitability in Poland demonstrated in their maps; in contrast, greater ecological suitability for CCHFV is seen in southern Russia and Kazakhstan. This is likely to be due to the different set of covariates used in each set of models (for example, Okely et al. used a Principal Components Analysis to reduce all available bioclimatic variables to a small number of factors, whilst our models selected those considered most important according to the literature, as well as a measure of shrub land cover). It should also be noted that Okely et al. do not incorporate the distributions of the vectors, which are not found in Poland. It is worth noting that our unmasked map of CCHF suitability (bottom right panel of Figure 5) also suggests many areas (e.g., Scotland) to be environmentally suitable for CCHF occurrence that are far beyond recorded occurrence locations and the predicted range of the two vector species. This underlines the need for vector masking to produce a realistic disease prediction but also suggests that if the vectors spread to these areas, the disease might as well.

Whilst the predicted vector and vector-masked disease outputs reflect the known data well for most of the area of interest, the recorded (and unmasked predicted) disease occurrence in certain parts of Western Asia is probably incorrectly masked by the vector distribution mask. The literature reports *H. marginatum* to be present (widespread) in Iran [13,34,35], further south than the distribution range boundary predicted by Kolonin [13], so it is likely that the point data available are substantially underrepresented for these regions (Iraq and Iran are not inside the region covered by VectorNet) and also perhaps some of the CCHF pseudo-absences were incorrectly placed in suitable areas in this region. Otherwise, it may be that the vector (and thus the virus in vectors) is present in pockets (perhaps of irrigation) that these spatial suitability models do not ‘detect’.

Also, a number of other *Hyalomma* species including *H. impeltatum*, *H. anatolicum*, *H. asiaticum*, *H. excavatum,* and *H. dromedarii* have been identified as CCHFV vectors in Iran and its neighbours [13,36]. These are not present in Europe (and there are few if any geo-referenced occurrence records) and were therefore not incorporated into the vector masking for Europe, but could improve the prediction in Western Asia. If so, this suggests that the unmasked disease models are more reliable in such areas.

A further potential improvement would be masking the maps according to the presence of host species for ticks, such as deer (for *H. marginatum*), hares, and wild rabbits (for *H. lusitanicum*), and limiting the surveillance priority to within rural areas and/or those that are nearer to livestock populations.

The updated and improved maps presented in this manuscript could serve as a starting point for a wider discussion about the potential future likelihood of occurrence of CCHF in Europe. Increasing awareness in public health experts and the medical profession in geographic areas, which we have newly shown to have a high probability of disease occurrence, may lead to CCHF being considered in the differential diagnosis of people with appropriate symptoms and a history of tick bite, and possibly better diagnoses in those areas. Future serological studies focussed in areas identified as high-risk in this manuscript could also lead to additional occurrence records that would in turn improve models. As evidenced by reports of the disease occurring in Spain in 2016 (before which a 2015 publication did not identify this as a suitable CCHFV transmission area), it is important for the mapping of high-suitability areas to be updated regularly, taking into consideration advancements in understanding of the disease distribution, any climate-related shift in vector distributions, as well as new covariate data sources.

## Figures and Tables

**Figure 1 insects-14-00771-f001:**
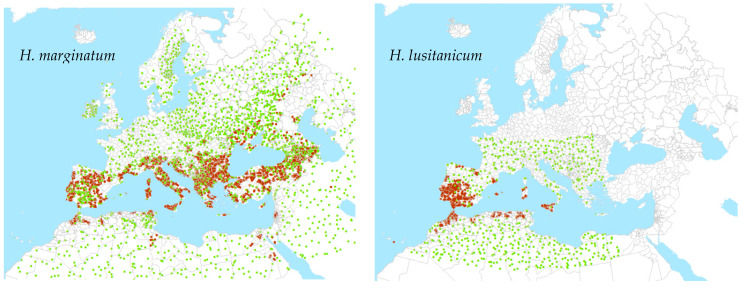
Points used for modelling of tick vectors. Brown is presence; green is absence.

**Figure 2 insects-14-00771-f002:**
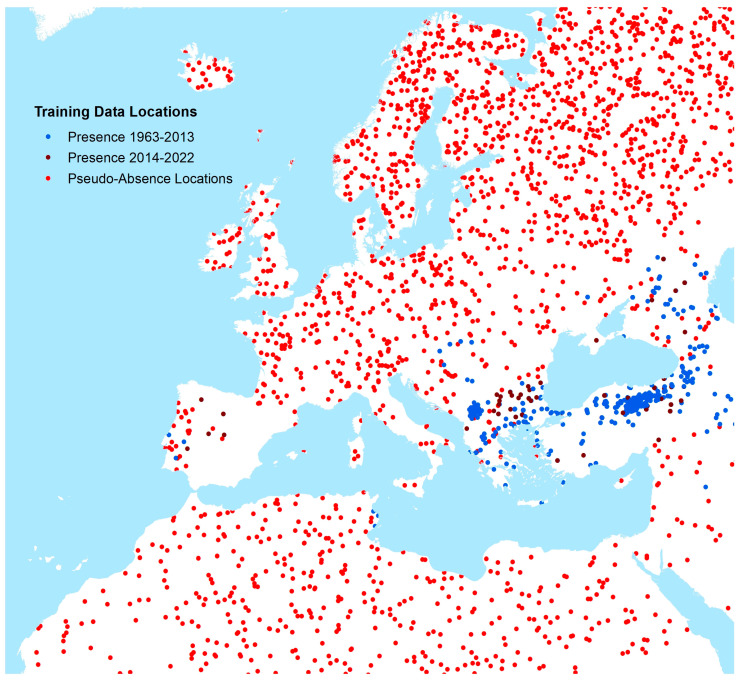
Occurrence and pseudo-absence locations of CCHF human cases (Europe and neighbouring areas).

**Figure 3 insects-14-00771-f003:**
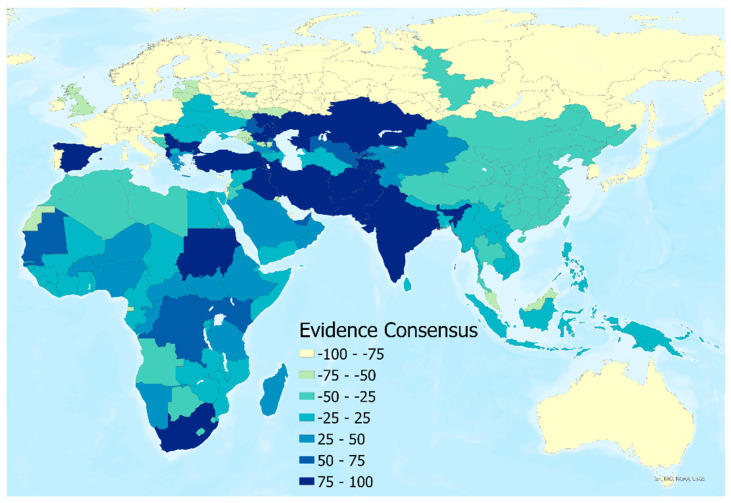
Evidence consensus map for human CCHF presence or absence by country.

**Figure 4 insects-14-00771-f004:**
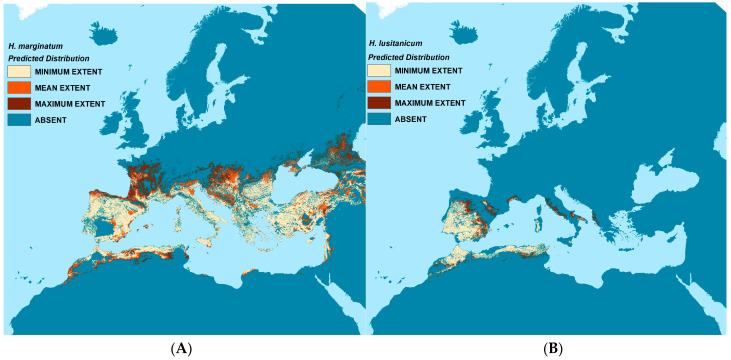
Predicted vector distribution maps. Left (**A**): *H. marginatum*; Right (**B**): *H. lusitanicum*.

**Figure 5 insects-14-00771-f005:**
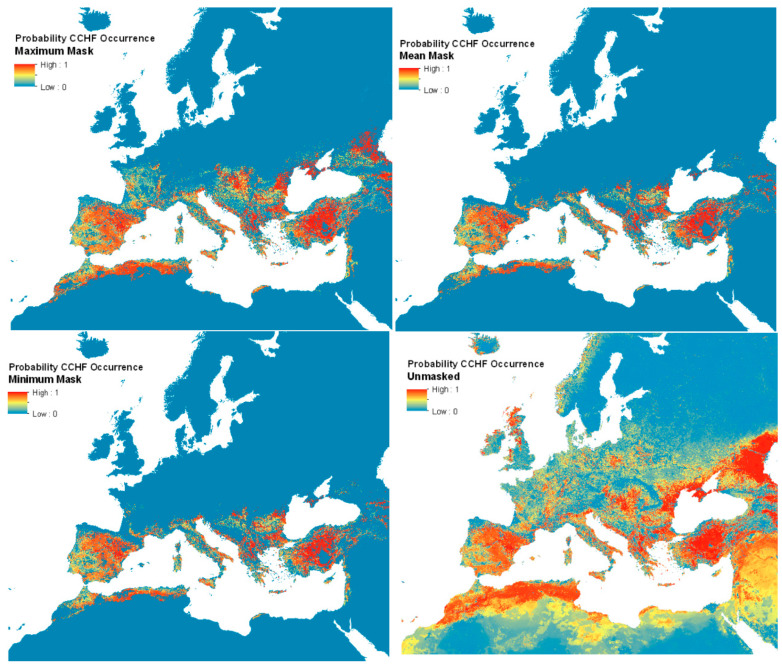
Predicted human CCHF suitability maps.

**Figure 6 insects-14-00771-f006:**
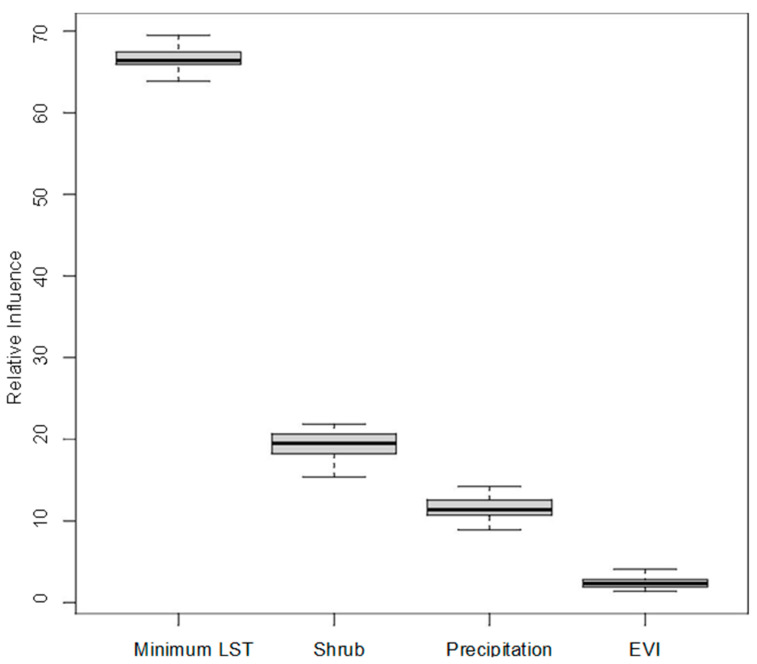
Proportional influence of covariates on CCHF suitability predictions.

**Figure 7 insects-14-00771-f007:**
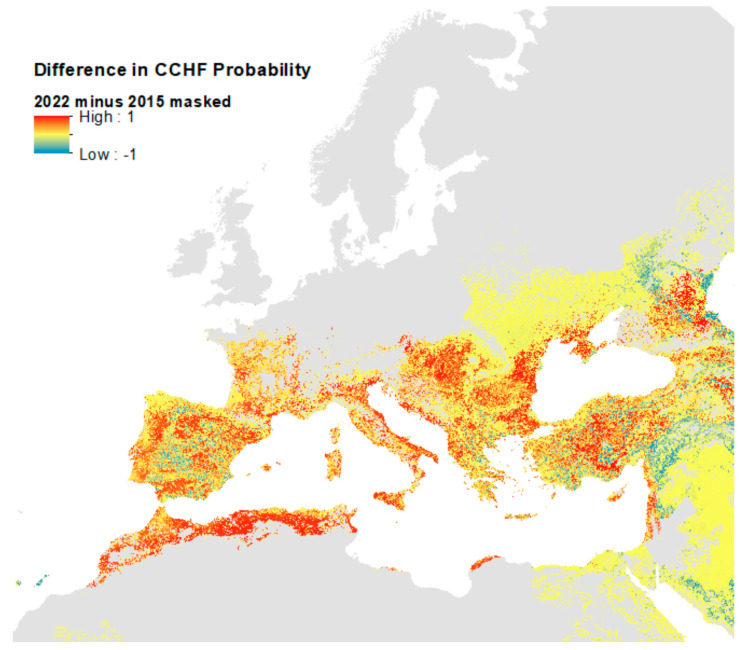
Difference in probability of CCHF suitability predictions between 2015 and 2022 models.

**Figure 8 insects-14-00771-f008:**
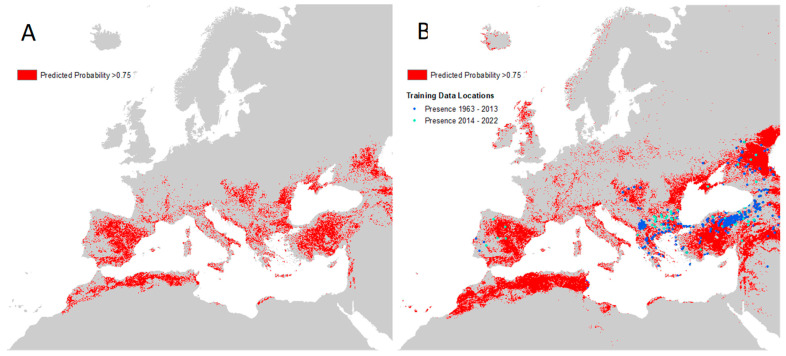
Maps of high predicted CCHF suitability. (**A**) Left: areas with predicted probability of disease ≥0.75 masked to maximum predicted vector extent; (**B**) Right: same as left but with recorded occurrence points and no vector mask.

## Data Availability

All spatial data inputs and outputs are provided as two ARCGIS 10.8 Packages: (1) ‘VNEREGOCCHF_Vector_Report_VectorCCHFData_April23.mpk’ containing the input and output Vector and CCHF Datasets; and (2) VNEREGOCCHF_Vector_Report_CovariateData_April23.mpk. The CCHF data locations are also provided in an excel file ‘CCHF Locations.xlsx’. The files can be downloaded from the following link: https://tinyurl.com/VNCCHFDATAFEB23, accessed 10 September 2023. Individual file descriptions are given in the package’s Table of Contents, and the associated filenames can be obtained from the Layer Properties Source Table. An endnote library (CCHF_2013_2022.xml) provides the references for all the literature searched for CCHF occurrences, and a second file (cchf_report_2023_final.xml) provides all references cited in the manuscript.

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
