# Peer review of "The Spatial Distribution of Crimean–Congo Haemorrhagic Fever and Its Potential Vectors in Europe and Beyond"

_insects, 2023, doi:10.3390/insects14090771_

Round 1
Reviewer 1 Report
The study constitutes the extension of a previous one (2015) on the prediction of CCHF distribution and its vectors. We suggest the modification of the current title as it misleads about the subject matter of the text. The authors should take into account that H. lusitanicum cannot be considered a vector, only a potential vector, as its vectorial capacity and competence have not yet been demonstrated. The study is correct and the results are coherent, although, in my opinion, some parameters such as the host population (especially for H. lusitanicum) should have been included for a more adequate prediction in accordance with the field results. Improvements are suggested for the understanding of some figures and the adaptation of species names to the scientific standard.

Author Response
Review 1
Thank you for your comments – well received
Title
We suggest the modification of the current title as it misleads about the subject matter of the text
Author:WW Subject:Note Date:01/09/2023 11:21:33
Please see also Reviewer 3 comments about tile
We have changed the title to “The spatial distribution of Crimean-Congo haemorrhagic fever and its potential vectors in Europe and beyond”
Page 1
Author: Subject:Nota adhesiva Date:30/08/2023 02:09:31
Author: Reviewer 1 Date:30/08/2023 02:09:31
.
Hyalomma marginatum and Hyalomma lusitanicum (in italics and by the first time the name of the species sould complete including genus)
Vectorial competence of H. lusitanicum has not yet been demonstrated cannot be said to be a vector, only that viral RNA has been detected in this species.
Author:WW Subject:Note Date:01/09/2023 11:21:33
latin names Italicised, and text modified to read "known and potential vectors"
Author: Reviewer 1 Date:30/08/2023 02:09:31
Italics
Author:WW Subject:Note Date:01/09/2023 11:22:03
Italiicised
Author: Reviewer 1 Date:30/08/2023 02:09:31
Italics
Author:WW Subject:Note Date:01/09/2023 11:22:03
Italiicised
Page:2
Author: Subject:Nota adhesiva Date:30/08/2023 02:09:31
Author: Reviewer 1 Date:30/08/2023 02:09:31
.
Hyalomma marginatum and Hyalomma lusitanicum (in italics and by the first time the name of the species sould complete including genus)
Vectorial competence of H. lusitanicum has not yet been demonstrated cannot be said to be a vector, only that viral RNA has been detected in this species.
Author:WW Subject:Note Date:01/09/2023 11:21:33
latin names Italicised, and text modified to read "known and potential vectors"
Author: Reviewer 1 Date:30/08/2023 02:09:31
Italics
Author:WW Subject:Note Date:01/09/2023 11:22:03
Italiicised
Author: Reviewer 1 Date:30/08/2023 02:09:31
Italics
Author:WW Subject:Note Date:01/09/2023 11:22:03
Italiicised
Page:6
Author: Reviewer 1 Date:30/08/2023 02:09:31
Punctuate propperly the figure caption.
The figure legend says "Presence" and then includes "absence" as one of the parameters and "extent" in the other three. It is very confusing. I think it would be more appropriate to put in the legend Distribution of H. marginatum or H. lusitanicum and in the parameters "absence", low presence, medium presence and high presence. The colors chosen do not allow to appreciate the difference between the values of "extent", it would be very useful to change the color palette.
Author:WW Subject:Note Date:01/09/2023 11:30:47
legend and colours changed as suggested
Author: Reviewer 1 Date:30/08/2023 02:09:31
Italics
Author:WW Subject:Note Date:01/09/2023 11:28:15
Italicised
Page:10
Author: Reviewer 1 Date:30/08/2023 02:09:31
Italics
Author:WW Subject:Note Date:01/09/2023 11:32:32
Italicised
Page:11
Author: Reviewer 1 Date:30/08/2023 02:09:31
Italics
Author:WW Subject:Note Date:01/09/2023 11:33:19
Italicised
Author:WW Subject:Note Date:01/09/2023 11:34:24
Author: Reviewer 1 Date:30/08/2023 02:09:31
No just hares, mainly wild rabbits. This is the main point for the population of H. lusitanicum, the abundance of rabbits. In my oppinion should be included
in the prediction.
Author:WW Subject:Note Date:01/09/2023 11:36:05
text changed to "hares and wild rabbits" added
Author:WW Subject:Note Date:01/09/2023 12:01:34
We did consider using hosts as a mask, but discarded the idea as generating a host mask for H marginatum would have involved many species (deer, livestock, small mammals, passerines) for some of which pixel resolution distribution data are not reliably obtainable (especially for the birds). For H. lusitanicum both recent GBIF records (see below for hares and rabbits) and published distributions from e.g. Estrada-Peña and colleagues suggest that a combined hare and rabbit layer would have covered the great majority of the vector range with very few gaps (see below) GBIF occurrence maps for hares and rabitts and so masked out very little of the tick distribution. We also hoped that incorporating land cover land use masks for vectors compensated for the h]lack of a host mask
GBIF occurrence records Hares and wild Rabbit

Reviewer 2 Report
I find the paper interesting and well-written. I have some minor comments. Regards.

Author Response
Thank you for your comments
We have implemented all the minor corrections and figure changes requested.

Reviewer 3 Report
The spatial distribution of Crimean-Congo haemorrhagic fever and its vectors in Europe and neighbouring areas
SPECIFIC TEXT REVIEW NOTES:
Line 13- change if to of
Line 47 -“are at greater risk at risk and fatality” to be re written for clarity
Line 81 -Study locality seems to include North Africa and yet the title of the article is restricted to Europe.
Line 112- change the word though to through
Line 126- replace to with by
Line 167- 170 The term “pseudoabsence” is mentioned in the article but not reflected in the results
Line 301- Other potential vectors of CCHFV are casually mentioned; these should be mentioned in the introduction.
GENERAL OBSERVATIONS :
§ The paper has highlighted new information with respect to risk areas for CCHFV which is vital for the health and related sectors. This is a significant and commendable contribution.
§ Results are presented for localities beyond Europe ( e.g figure 4); does this call for revision of the title of the paper?
§ The Literature survey is impressive and quite extensive but the existing geographic distribution of CCHFV by W.H.O, which are available, are not mentioned or referenced in the article.
Minor editing of English language required
Author Response
Thank you for your comments
SPECIFIC TEXT REVIEW NOTES: We have now addressed all of these minor changes.
Note re: Line 167- 170 The term “pseudoabsence” is mentioned in the article but not reflected in the results – The pseudo absence locations are generated as part of the modelling procedure and as such, they only need to be referenced in the methods section and are displayed in Figure 2.
GENERAL OBSERVATIONS :
- The paper has highlighted new information with respect to risk areas for CCHFV which is vital for the health and related sectors. This is a significant and commendable contribution.
- Results are presented for localities beyond Europe ( e.g figure 4); does this call for revision of the title of the paper?
See also Reviewer 1c comments
The title has now been changed to: “The spatial distribution of Crimean-Congo haemorrhagic fever and its potential vectors in Europe and beyond”
- The Literature survey is impressive and quite extensive but the existing geographic distribution of CCHFV by W.H.O, which are available, are not mentioned or referenced in the article.
As far as we can see, the WHO does not provide an updated map of the global distribution of CCHF. Can the reviewer provide a link?
Round 2
Reviewer 3 Report
The paper is a an updated contribution to distribution of Crimean - Congo haemorrahgic fever and its vectors in Europe and beyond.